# Revisiting Feature Interaction Selection in Neural Additive Models

## Abstract

In this work, we revisit the paradigm of feature interaction selection for additive models. This paradigm generalizes the selection of a model's input features to the selection of a model's feature interactions by equipping any model with the additive structure of a generalized additive model. When applied to neural networks, this restricts the network's learned representations to interactions between the specified sets of features. In the study of the training dynamics of these neural additive models, we discover a new phenomenon which we call 'medium dimensionality', corresponding to a balance between data complexity and model complexity. We find that this phenomenon helps to explain the good performance of additive models on tabular datasets. We moreover find that the tool of additive models is able to unify insights for many of the recently explored phenomenon of deep learning theory: double descent, grokking, leap dynamics, and the staircase property. Finally, we present developments on selections algorithms and neural additive models, benchmarking performance across a suite of tabular datasets.

## 1 Introduction

The impressive performance of deep neural networks over the past decade has inspired countless investigations into *what* representations they learn and *how* they learn these representations.

A large body of work, now broadly called *deep learning theory*, seeks to understand how and what these neural networks learn from features starting from first principles. This lofty goal is balanced with a healthy amount of empirical studies describing the many unexpected phenomenon of training with deep neural networks. Beginning with works which questioned the classical statistical foundations in deep learning applications (Zhang et al., 2017), new descriptions provide toy models which follow observed phenomenon like double descent (Belkin et al., 2019) and grokking (Power et al., 2022). Recent works have focused on regimes like sparse feature learning and sparse polynomial learning as powerful toy models (Woodworth et al., 2020; Telgarsky, 2023; Suzuki et al., 2023).

Somewhat similarly, the area of *interpretability* attempts to explain deep learning, but from the opposite direction of explaining what a neural network or blackbox model has already learned. There have been many approaches to explainability and interpretability over the years (Rudin, 2019). Although mechanistic interpretability still dominates the study of transformer models (Schubert et al., 2020; Olah et al., 2020), the model-agnostic research of interpretability has developed entirely different tools for understanding machine learning and deep learning black boxes. Major approaches in this area include feature attribution methods like Integrated Gradients (Sundararajan et al., 2017) and SHAP (Lundberg & Lee, 2017), as well as the dual notion of interpretable models (Caruana et al., 2015; Chang et al., 2021). Generalized additive models have greater interpretability and robustness than blackbox algorithms and have recently been shown to match state-of-the-art performance on tabular datasets despite their simplified form (Chang et al., 2022; Enouen & Liu, 2022).

In this work, we revisit why a simplified additive model is able to match or outperform blackbox methods, finding that the statistical robustness of these simplified models is a key element leading to their success. We call this property medium-dimensionality and explore how additive models can be used as a tool to help illustrate the effects of statistical complexity both in machine learning and deep learning. We demonstrate how many known deep learning phenomenon also apply to neural additive models and revisit these many phenomenon using a data-centric lens.

Our contributions are as follows: (1) highlighting of the 'medium dimensionality' property which balances model complexity and statistical complexity at realistic scales; (2) using additive models as a tool to better understand the statistical effects underpinning existing deep learning phenomenon; (3) further developing the state-of-the-art techniques for feature interaction selection in neural additive models; and (4) empirical demonstration of GAMs matching state-of-the-art performance on a wide array of tabular datasets. In Section 2, we provide the necessary background on deep learning theory and generalized additive models. In Section 3, we extend existing deep learning theory using the tool of neural additive models for data-centric insight. In Section 4, we provide practical improvements on the algorithms used for training neural additive models. In Section 5, we combine theoretical support and practical developments to benchmark the good performance of neural additive models on various tabular datasets.

## 2 BACKGROUND

Let $n, d, c \in \mathbb{N}$ be the number of samples, input dimension, and output dimension respectively. We write $x \in \mathcal{X} \subseteq \mathbb{R}^d$ and $y \in \mathcal{Y} \subseteq \mathbb{R}^c$ where $d$ is the number of input features and $c$ is the number of output dimensions (number of classes in the case of classification). We focus on the two central machine learning tasks of classification and regression. In our experiments, $c$ will always be 1 in the case of regression. Broadly, our goal will to learn an approximation function $y \equiv f(x)$. We will also write a set of features (called a feature interaction) as $S \subseteq [d] := \{1, \ldots, d\}$. We write the powerset as $\mathcal{P}([d]) := \{S : S \subseteq [d]\} \cong \{0, 1\}^d$. We will write $x_S = (x_i)_{i \in S}$ and $x_{-S} = (x_j)_{j \in ([d]-S)}$.

### 2.1 DEEP LEARNING THEORY

Deep learning theory has developed to understand how a network's learned representations and training dynamics are shaped by choices of the deep learning architecture. This field generally takes the form of being guided by empirical phenomenon while constantly making progress towards a theoretical explanation of the phenomenon. We list several key developments which are relevant for our discussion. We will later discuss how the usual model-centric and training-centric lens taken for these empirical phenomenon can be enhanced by a data-centric view highlighting the important role which training data plays, see Figure 1.

**Double Descent** The phenomenon of double descent (Belkin et al., 2019) describes how the goodness-of-fit for a statistical model behaves as its number of parameters increases, finding a local minimum in the underfitting regime with a small number of parameters and another minimum at the limit of infinite parameters. Later verified on deep neural networks (Nakkiran et al., 2020), this phenomenon has been used to explain the large empirical success of deep neural networks despite their overparametrization. This was later characterized in greater detail with the infinite variance spike occurring right at the interpolation point (Hastie et al., 2020) and kicked off research into the related ideas of 'benign overfitting' (Bartlett et al., 2020).

**Grokking** Another deep learning phenomenon is that of grokking which is something of an 'almost-benign' overfitting, where the neural network first overfits to the training data, but after some delay the network learns to generalize perfectly (Power et al., 2022). Follow up works demonstrated the same delayed learning behavior on real-world datasets, finding evidence of two learning phases: one for overfitting and one for weight decay (Liu et al., 2023). Later works embed this weight decay mechanism into the literature on rich feature training (Kumar et al., 2024) and some works have even suggested numerical stability is the main culprit (Prieto et al., 2025).

**Single-Index Model** Much of the interest in a theory of deep learning revolves around the most exciting property of neural networks, their ability to automatically learn features from the data. This generated much interest in explaining network behavior beyond the lazy NTK regime and in rich feature-learning dynamics (Woodworth et al., 2020; Li et al., 2021). One well-studied toy model is the single-index and multi-index models, which construct a small set of $p$ linear features which are then nonlinearly combined with arbitrary complexity, $f(x) = g(\sum_i \beta_i^1 x_i, \ldots, \sum_i \beta_i^p x_i)$ (Bietti et al., 2022; Damian et al., 2024) This is a toy model of feature learning[1] with simple linear features. We will later discuss how this research is complementary to the additive model.

---

[1]Note the distinction between features as in *learned features* (i.e. representations) and features as in *input features* (which will become centerstage in the discussion on additive models).

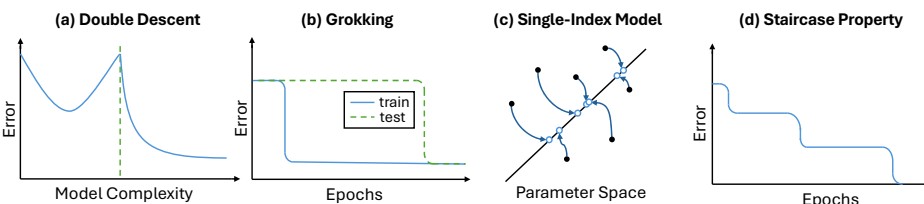

Figure 1: Simplified illustrations of several deep learning phenomena.

**Staircase Property** The staircase property describes how during typical gradient dynamics of SGD, the learned model tends to learn functions of increasing complexity, first demonstrated on linear and nonlinear classification (Kalimeris et al., 2019) and later explored in much greater detail on sparse polynomials (Abbe et al., 2021; 2022). The name describes how the model climbs up the hierarchical structure of complexity or how the error climbs down the stairs of plateaus corresponding to each complexity level. More recent works have characterized the learning behavior in greater detail (Abbe et al., 2023) and demonstrated similar properties across a wide array of specialized two-layer architectures (Kunin et al., 2025). We will later show how the additive model is yet another wide class of examples obeying the staircase property for neural networks.

## 2.2 GENERALIZED ADDITIVE MODELS

The generalized additive model (GAM) of Hastie & Tibshirani (1990) is a functional form which can be equipped to any machine learning methods, learning a (nonlinear) 1D relationship in each of the $d$ variables jointly while maintaining a simple additive structure for the overall model. Modern study centers around its interpretability benefits due to the simple model structure (Lou et al., 2012).

$$F^{\leq 1}(x_1, \ldots, x_d) = f_\emptyset + f_1(x_1) + \cdots + f_d(x_d) \tag{1}$$

This idea can be easily generalized to include two-dimensional interactions effects in addition to the one-dimensional main effects (Wahba et al., 1994). This model class is still often seen as an interpretable model because its 2D functions can still be visualized using a heatmap plot (Lou et al., 2013; Yang et al., 2020; Chang et al., 2022).

$$F^{\leq 2}(x_1, \ldots, x_d) = f_\emptyset + f_1(x_1) + \cdots + f_d(x_d) + f_{1,2}(x_1, x_2) + \cdots + f_{d-1,d}(x_{d-1}, x_d) \tag{2}$$

In recent years, modern machine learning methods have added higher-order interaction effects, pushing beyond the quadratic complexity of learning 2D additive models to achieve even better performance on certain datasets (Yang et al., 2020; Dubey et al., 2022; Enouen & Liu, 2022). For some order $k \geq 3$, we may define the higher-order GAM-k as:

$$F^{\leq k}(x_1, \ldots, x_d) = f_\emptyset + \sum_{i \in [d]} f_i(x_i) + \ldots + \sum_{S \subseteq [d], |S|=k} f_S(x_S) = \sum_{S \in \mathcal{I}_{\leq k}} f_S(x_S) \tag{3}$$

where we write $\mathcal{I}_{\leq k} := \{S \subseteq [d] : |S| \leq k\}$. Unfortunately, plotting a 3D function is not quite as feasible and so these models lose some of their interpretability guarantees. Nevertheless, as we will later discuss, they maintain robustness guarantees due to their simpler functional form.

## 2.3 FEATURE INTERACTION SELECTION

Although feature attribution and feature selection evolved independently from additive models, the study of feature interactions is intimately connected with additive models.

**Definition 1.** A $k$-dimensional, non-additive **feature interaction** between features $\{s_1, \ldots, s_k\} = S \subseteq [d]$ *does not exist* in the function $f : \mathbb{R}^d \to \mathbb{R}$ if we can write:

$$f(x) = f_{-s_1}(x_{-s_1}) + \cdots + f_{-s_k}(x_{-s_k}), \tag{4}$$

for some functions $f_{-s_1}, \ldots, f_{-s_k}$. We say that the interaction does not exist here because we wrote $f(x)$ as an additive model which never used a term containing $S$. We can make this explicit via the equivalent condition: $f(x) = \sum_{\{T \subseteq [d] \text{ s.t. } S \not\subseteq T\}} f_T(x_T)$.

**Feature Attribution**  Feature attribution remains the dominant paradigm for the study of explainability in machine learning, with the stated goal of: 'determine how the input features affected the output classification or regression'. This usually means to provide an attribution $\phi_i(x)$ to each input feature $x_i$, and includes attribution methods like Integrated Gradients (Sundararajan et al., 2017) and SHAP (Lundberg & Lee, 2017). Although much research has been done on refining these methods, more recent developments have focused on providing interactions attributions as well, filling in the gaps where feature attribution alone is not sufficient to explain the model predictions. This includes methods like Shapley-Taylor (Sundararajan et al., 2020) and Archipelago (Tsang et al., 2020). These modern approaches generalize to interactions by providing an attribution $\Phi_S(x)$ to each subset of features $x_S$, allowing greater nuance in the provided explanations.

**Feature Selection**  Feature selection has an even longer history than feature attribution, focusing on the question of: 'which features do I include in my statistical model?' Although this area is much too broad to review in any detail, a major related area of relevance is the study of high-dimensional statistics, which focuses on the regime when $d \gg n$. This includes classical feature selection for linear model like LASSO (Tibshirani, 1996) and OMP (Mallat & Zhang, 1993). Here sparsity in the form of feature selection is necessary due to insufficient samples to fit even the simple linear model. The first extensions to 'feature interaction selection' included the pairwise bilinear terms $\beta_{ij}x_ix_j$ in addition to the linear terms $\beta_i x_i$ and performed selection over all of the pairs (Bien et al., 2013; Hao & Zhang, 2014; Fan et al., 2016). Here we may imagine the regime of $d^2 \gg n$ instead because of the quadratic number of pairs. Parallel developments applied the same statistical reasoning to nonlinear additive models like SpAM (Ravikumar et al., 2009) and COSSO (Lin & Zhang, 2006).

It was not until later that feature interaction selection was described in full generality as a problem of selecting from all possible higher-order interactions, choosing an entire collection $\mathcal{I} \subseteq \mathcal{P}([d])$ from all $2^d$ possible feature subsets (Sugiyama & Borgwardt, 2019; Enouen & Liu, 2022). Unlike high-dimensional statistics, which is mainly applied to specific domains like biostatistics (where $d \gg n$ often holds), feature interaction selection is more widely applicable across machine learning tasks (since $2^d \gg n$ more commonly holds), especially in the ever-increasing presence of high-dimensional data[2]. It is this property which we will call medium-dimensional statistics or **medium dimensionality**. Even for relatively small datasets with $d = 20$ or $d = 30$, we quickly face statistical limitations due to the fact that $2^d = 1.0e6$ or $2^d = 1.1e9$ is often much larger than $n$, the sample size.

## 3 EMPIRICAL THEORY

We revisit many of the empirical phenomenon studied in the deep learning theory from a statistical perspective of additive models and medium-dimensional statistics. Although much progress has been made on studying these empirical phenomena through the lens of model complexity (e.g. number of parameters) and algorithm complexity (e.g. number of gradient steps), it seems that the lens of data complexity is under appreciated in the current literature. We propose the additive model as a general tool which is able to characterize and study many of these same phenomenon when viewed from the sample complexity perspective.

### 3.1 MEDIUM DIMENSIONALITY

We first demonstrate that this statistical phenomenon does exist on a simple synthetic dataset. We generate simple synthetic data which obeys the additive model structure of order three, generating an arbitrary function for each interaction's shape function, normalizing appropriately. For additional details, see the appendix. In Figure 2a, we plot the final test performance of a neural additive model of order 1, 2, and 3 as we vary the number of training samples provided to the algorithm. On the LHS, we see all algorithm struggling to learn much from the small datasets. On the RHS, we see all algorithms achieve their theoretical best performance (which are equispaced due to data construction). Importantly, in a range of about 200 samples to about 2000 samples, the 'medium simplicity' GAM2 model is the best performing model.

---

[2]Note the distinction between *high-dimensional statistics* which has the meaning that $\{d \gg n\}$ and *high-dimensional data* which has the more colloquial meaning that $d$ is large i.e. $\{d \gg 1\}$.

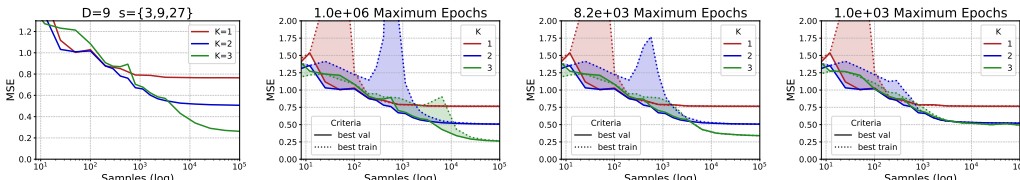

Figure 2: Full test error sweep across dataset size using dataset parameters $D = 9$ and $\vec{s} = (3, 9, 27)$. (a) Test error for three model classes across a variety of dataset sizes with early stopping according to best validation performance. (b) Double descent spikes at various locations for each of the three model types. (c, d) Double descent spikes with a greatly reduced computational budget, reducing the spikes considerably ($2^{20}$ steps reduced to $2^{13}$ and $2^{10}$).

## 3.2 DOUBLE DESCENT SPIKE LOCATION

In Figure 2b, we see the location of the double descent spikes for each of the different model classes. These each occur around the interpolation point for the GAM model. For neural additive models, these bad solutions appear at the end of training, where the neural network continues to try overfitting more and more to the data. Accordingly, we plot the test error for the best validation performance compared with the test error for the best training performance. In Figures 2c and 2d, we reduce the maximum number of training steps significantly, resulting in a complete disappearance of the double descent spikes. It is also worth noting that the powerful implicit bias of neural networks with gradient descent training followed by early stopping is able to greatly minimize these concerns of overfitting in practice as was seen in Figure 2a.

Although finding double descent spikes in these locations is not surprising, the mechanism by which neural additive models learn these bad minimum (long-run training) and the fact that the choice of interactions leads to this spike appearing at nearly any location are both important insights which are not easily gleaned without focusing on the dataset size as the variable of interest. Moreover, this helps illuminate that even for the exact same task and ground truth DGP (data generating process), only the statistical information of how many samples were observed can drastically change the learning behavior. Even if we know the true model, it may not be the best to fit with finite data.

## 3.3 TRANSITION BEHAVIOR

We first demonstrate grokking on a simple higher-order dataset of $f^{\text{XOR5}}(x) = x_1 x_2 x_3 x_4 x_5$ in $d = 10$ dimensions without noise. Additive models make it easy to construct grokking behavior by choosing a relatively high feature interaction and finding the dataset thresholds which leads to typical training and overfitting. In Figures 3a and 3b, we have discrete inputs ($x_i \in \{-1, 1\}$) and in Figures 3c and 3d, we have uniform inputs ($x_i \in [-\sqrt{3}, \sqrt{3}]$). Black corresponds to overfitting/grokking and orange corresponds to easy learning.

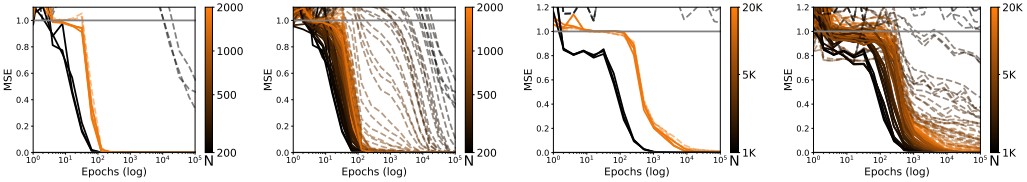

Figure 3: Transition from normal learning to grokking by varying the number of training samples. Colored by number of training samples. Solid line is training error, dashed line is validation error. From left-to-right: discrete XOR5 on the lowest and highest samples; discrete XOR5 on many sample values; continuous XOR5 on the lowest and highest samples; continuous XOR5 on many sample values.

In Figures 3b and 3d, we see that sweeping dataset size leads to a smooth interpolation of the validation curve from normal behavior to grokking behavior. Not only does the validation loss descend later, but training loss descends earlier. Although the typical mechanism of grokking is

through weight decay finding a simpler solution, these experiments help make clear that statistical complexity is the root cause, forcing the neural network to depend on its inductive biases.

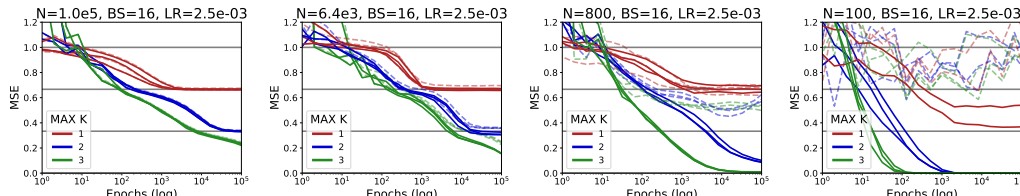

Figure 4: Staircase phenomenon without access to infinite data. Large datasets lead to a clean stairwise descent; however, reducing dataset size leads to a mixed phenomenon where the model learns some of the staircase steps but sometimes 'falls down' others, overfitting at that level and after.

We find that this behavior mimics the staircase phenomenon but for a single step. Research on the staircase training dynamics often operates under assumptions like large batch size or uncorrelated trajectories, leading to strong assumptions about how large the required dataset is (i.e. 'low dimensionality'). In reality, however, it may be of great interest to understand training dynamics in the medium-dimensional regime, where there is sufficient data for complex dynamics to emerge, but insufficient data for the population dynamics to take hold. Indeed, in Figure 4 we find a distinct behavior at lower sample sizes, corresponding to some levels of the staircase being learnable and others being too complex to fit accurately. This results in behavior which resembles the training curve 'falling down the stairs' where the training loss descends while validation loss rises.

Accordingly, we envision the transition from perfect fitting to delayed fitting to grokking to overfitting as all part of the same spectrum which describes how the validation curve 'swings out' as the number of samples decrease and the neural network is forced to rely more and more heavily on its inductive biases. We find that the additive models with higher-order interactions provide an extremely useful test bed for this behavior.

## 3.4 STABILITY UNDER HYPERPARAMETERS

In Figure 5, we see that the general shape of the learning curves is stable under variations in the learning rate and batch size, mostly dictating the 'speed' at which the model converges to the correct solution. This runs slightly counter to work looking at feature learning, which often focuses on carefully initialized small-scale weights at the edge of stability. We find that neural additive models are not particularly sensitive to these many changes and the main specification affecting learning behavior is the sample size and underlying additive structure.

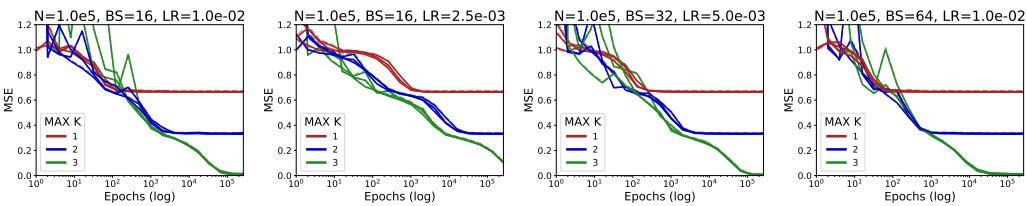

Figure 5: Large dataset training performance across different batch sizes and learning rates.

## 4 ALGORITHMIC DEVELOPMENTS

Given the importance of using neural additive models for better understanding deep learning theory in Section 3 and the empirical good performance of neural additive models which will be demonstrated in Section 5, we introduce three major improvements to the training of neural additive models. We focus on adjusting the training algorithm to effectively address the major shortcomings of neural additive models: the ease of higher-order interaction selection and the speed of learning.

**Algorithm 1** Layerwise Feature Interaction Selection (Enouen & Liu, 2022)

---

**Inputs**: Prediction model $f(x)$, and validation dataset $X^* = \{x^{(1)}, \dots, x^{(V)}\}$
**Parameters**: Maximum index $K$, heredity cutoff thresholds $\{\tau_k\}_{k=1}^{K}$, strength thresholds $\{\theta_k\}_{k=1}^{K}$, feature interaction explainer $\Phi_S(f; X^*)$
**Output**: $\mathcal{I}$, a family of feature interactions with index at most $K$

1: Set $\mathcal{I} \leftarrow \{\emptyset\}$
2: **for** $k = 1, \dots, K$ **do**
3:   $\mathcal{J} \leftarrow Candidates(\mathcal{I}, \{\tau_k\}_{k=1}^{K})$
4:   **for** $S$ in $\mathcal{J}$ **do**
5:     $\omega_S \leftarrow \Phi_S(f; X^*)$
6:   **end for**
7:   $\mathcal{K} \leftarrow \{S \in \mathcal{J} : \omega_S > \theta_k\}$
8:   $\mathcal{I} \leftarrow \mathcal{I} \cup \mathcal{K}$
9: **end for**
10: **return** $\mathcal{I}$

---

**Algorithm 2** Batchwise Feature Interaction Selection (Ours)

---

**Inputs**: Prediction model $f(x)$, and validation dataset $X^* = \{x^{(1)}, \dots, x^{(V)}\}$
**Parameters**: Maximum index $K$, heredity cutoff thresholds $\{\tau_k\}_{k=1}^{K}$, batch size $B$, total rounds $T$, feature interaction explainer $\Phi_S(f; X^*)$
**Output**: $\mathcal{I}$, a family of feature interactions with index at most $K$

1: Set $\mathcal{I} \leftarrow \{\emptyset\}$
2: **for** $t = 1, \dots, T$ **do**
3:   $\mathcal{J} \leftarrow Candidates(\mathcal{I}, \{\tau_k\}_{k=1}^{K})$
4:   **for** $S$ in $\mathcal{J}$ **do**
5:     $\omega_S \leftarrow \Phi_S(f; X^*)$
6:   **end for**
7:   $\mathcal{K} \leftarrow TopK(\mathcal{J}, \{\omega_S\}_{S \in \mathcal{J}}, B)$
8:   $\mathcal{I} \leftarrow \mathcal{I} \cup \mathcal{K}$
9: **end for**
10: **return** $\mathcal{I}$

---

## 4.1 BATCHWISE SELECTION

We propose to modify the feature interaction selection algorithm for neural networks originally proposed by Enouen & Liu (2022) in Algorithm 1. This algorithm uses an XAI feature interaction detection method on a blackbox model $\Phi_S$ on a candidate interaction $S$. Their work adds interactions to the selected set layer-by-layer threshold, using heredity ($\tau_k$) to limit the overall number of visited interactions from growing too quickly. We find two key limitations of this approach. First, the parameters $\theta_k$ are absolute parameters on the scale of the XAI measurements $\Phi_S$, making them difficult to tune. Second, the layerwise algorithm takes an 'all-or-nothing' approach for each of the interactions, measuring each candidate a single time and comparing it against this difficult-to-tune absolute threshold.

We propose Algorithm 2 to solve these challenges, using a batchwise approach with hyperparameters interactions per batch ($B$) and number of rounds ($T$). Although a simple change, it is quite powerful because the parameters $B$ and $T$ are much easier to tune: $B$ is set to a small fixed number and $T$ is run iteratively until model overfitting or computational timeout.

## 4.2 MASKED TRAINING

The next modification we make is to the XAI explainer method which computes $\Phi_S$. Inspired by modern SHAP approaches (Jethani et al., 2022), we find that recent XAI approaches use masked training to define a surrogate model for the explainer rather than post-hoc averaging the contributions over the marginal distribution. For greater details on the difference between marginal explanations and conditional explanations, see (Covert et al., 2021). For our purposes, the main benefit is the need to query the model significantly fewer times when computing the interaction score in $\Phi_S$ from Algorithm 1 or 2. We can see in Figure 6 the large reduction in time taken for this version of the

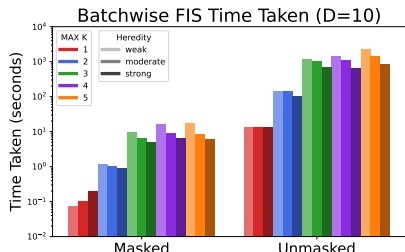
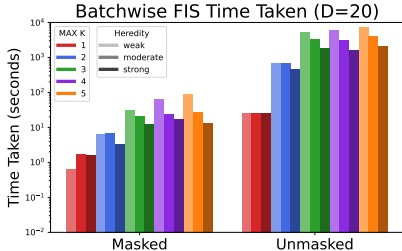

Figure 6: Time taken to complete the FIS algorithm when using masked models or when using unmasked models. Various choices of heredity and $K$ value are plotted.

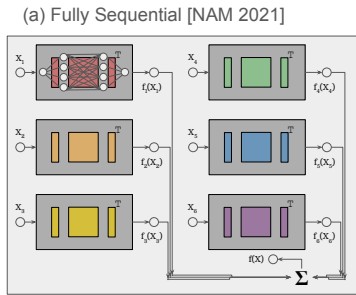 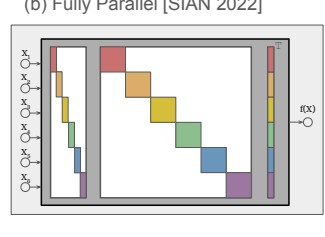 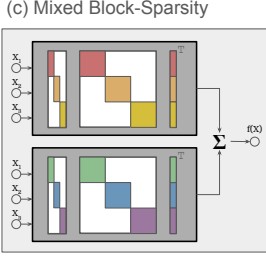

Figure 7: Various equivalent implementations of a neural network which respect the additive model structural equation. Feature interactions are not depicted, but only require a simple change to the first input layers. (a) A list of smaller networks are each called sequentially across the summation to compute the one-dimensional shape functions. (b) Each weight matrix is embedded inside of a larger blocksparse matrix to allow for much greater parallelism on a GPU. (c) An in-between implementation which only partially inflates the model.

algorithm. Because this score is called often throughout the interaction selection, scaling to higher-order interactions and larger numbers of interactions benefits greatly from these changes.

### 4.3 MIXED BLOCK SPARSITY

Another key development is in the architectural specification of the neural additive model. The natural implementation of an additive model using neural networks is to take several smaller sub-networks which each represent one of the shape functions corresponding to an interaction (Agarwal et al., 2021). Unfortunately, this naive implementation does not scale well with the parallelism of the GPU, becoming prohibitively slow with even a moderate number of interactions. Later work proposed to completely parallelize the weight matrix computation by embedding the subnetwork matrices into a larger blocksparse matrix which emulates these many smaller networks (Enouen & Liu, 2022); however, with enough interactions it becomes too large to even fit on the GPU.

In this work, we propose to balance between these two extremes. Instead of inflating the subnetworks to the maximally sized blocksparse matrix, we allow for partial inflation up to a specified maximum size. This allows for better GPU utilization without facing the quadratic memory requirements of a fully blocksparse model. All three approaches are depicted in Figure 7.

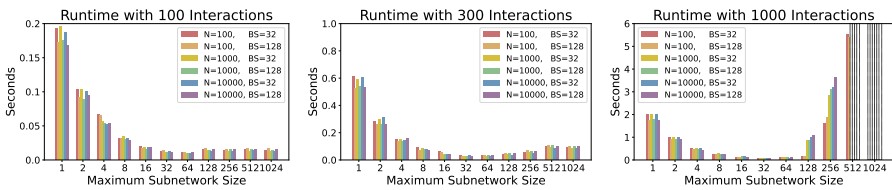

Figure 8: Average runtime for a single step of gradient descent. Each plot corresponds to a choice of number of interactions for the neural additive model (the main factor affecting network size). Multiple choices of batch size and full dataset size are plotted in different colors. Networks larger than the 16GB of GPU memory are depicted with a white bar. Max size 1 corresponds to Figure 7a. Max size 1024 corresponds to Figure 7b.

In Figure 8, we benchmark the time taken for networks interpolating between NAM and SIAN. At the beginning, we see an exponential decrease in the time taken as we double the subnetwork size, corresponding to the network more efficiently utilizing the GPU parallelism. However, this decrease does not continue forever, becoming less efficient after combining around 32 64 interactions. This happens even before the GPU is completely out of memory. This behavior is explained by the efficient usage of GPU cores being balanced with the need to balance GPU memory. Once the large, inflated networks become too much work to move around the GPU SRAM, the larger networks become a burden of memory management instead of more effectively using GPU cores. Interestingly, this means that our proposed approach in Figure 7c is both faster and more space-efficient than the solution proposed in (Enouen & Liu, 2022).

## 5 REAL-WORLD EXPERIMENTS

We compare across a suite of 16 datasets and against a large array of blackbox machine learning methods and interpretable additive model methods. Additional dataset and model details are in the appendix.

Table 1: Test metrics on the classification datasets (ROC AUC), higher is better.

| Model | Madelon (⇑) | Bankruptcy (⇑) | Higgs Boson (⇑) | Adults (⇑) | Tree Cover (⇑) | Credit Card Fraud (⇑) | Eucalyptus (⇑) | Customer Churn (⇑) |
|---|---|---|---|---|---|---|---|---|
| LASSO | 0.588±0.016 | 0.896±0.016 | 0.684±0.000 | 0.904±0.000 | 0.921±0.000 | 0.984±0.003 | 0.856±0.012 | 0.847±0.002 |
| LASSO† | 0.609±0.011 | 0.884±0.027 | 0.684±0.000 | 0.904±0.001 | 0.921±0.000 | 0.984±0.001 | 0.852±0.014 | 0.847±0.001 |
| GA$^2$M EBM | 0.844±0.013 | 0.937±0.002 | 0.808±0.000 | 0.927±0.001 | 0.943±0.000 | 0.991±0.002 | 0.908±0.008 | 0.845±0.002 |
| GA$^2$M EBM† | 0.878±0.017 | 0.939±0.003 | TLE | **0.928 ± 0.001** | 0.943±0.001 | 0.990±0.002 | 0.910±0.006 | 0.848±0.000 |
| NODE-GAM | 0.570±0.022 | **0.964±0.001** | 0.730±0.001 | 0.917±0.003 | 0.730±0.001 | **0.999±0.000** | **1.000±0.000** | 0.810±0.003 |
| NODE-GA$^2$M | 0.531±0.010 | 0.962±0.003 | 0.701±0.000 | 0.916±0.001 | 0.765±0.004 | **0.999±0.000** | **1.000±0.000** | 0.803±0.005 |
| SIAN-1* | 0.641±0.008 | 0.935±0.001 | 0.768±0.001 | 0.911±0.001 | 0.936±0.000 | 0.972±0.011 | 0.925±0.001 | **0.855±0.001** |
| SIAN-2* | 0.582±0.015 | 0.938±0.001 | 0.794±0.000 | 0.913±0.001 | 0.947±0.001 | 0.982±0.004 | 0.931±0.006 | 0.845±0.001 |
| SIAN-3* | 0.573±0.012 | 0.938±0.001 | TLE | 0.915±0.001 | 0.953±0.002 | 0.969±0.004 | 0.925±0.008 | 0.841±0.004 |
| SIAN-5* | TLE | 0.938±0.000 | TLE | 0.914±0.001 | 0.943±0.001 | 0.972±0.005 | 0.843±0.027 | 0.839±0.002 |
| MLP | 0.688±0.012 | 0.899±0.001 | 0.809±0.001 | 0.916±0.001 | 0.957±0.001 | 0.996±0.000 | 0.936±0.008 | 0.847±0.003 |
| RF | 0.707±0.010 | 0.915±0.009 | 0.834±0.000 | 0.902±0.002 | 0.997±0.000 | 0.962±0.008 | 0.915±0.008 | 0.827±0.004 |
| RF† | 0.797±0.004 | 0.929±0.007 | TLE | 0.917±0.001 | 0.996±0.001 | 0.984±0.006 | 0.911±0.009 | 0.850±0.002 |
| SVM | 0.617±0.010 | 0.866±0.009 | TLE | 0.899±0.001 | 0.966±0.000 | 0.960±0.014 | 0.917±0.004 | 0.807±0.004 |
| SVM† | 0.600±0.010 | 0.826±0.036 | TLE | 0.906±0.000 | TLE | 0.971±0.014 | 0.915±0.004 | 0.845±0.002 |
| XGB | 0.831±0.017 | 0.921±0.009 | 0.823±0.001 | 0.925±0.001 | 0.985±0.000 | 0.989±0.003 | 0.902±0.011 | 0.816±0.001 |
| XGB† | 0.865±0.017 | 0.933±0.004 | **0.856±0.000** | 0.927±0.001 | **0.999±0.001** | 0.992±0.002 | 0.913±0.006 | 0.846±0.002 |
| CatBoost | **0.888±0.005** | 0.932±0.006 | 0.800±0.000 | 0.923±0.000 | 0.961±0.000 | 0.993±0.002 | 0.915±0.008 | 0.834±0.013 |
| CatBoost† | 0.881±0.013 | 0.934±0.002 | 0.842±0.002 | 0.928±0.000 | 0.994±0.001 | 0.989±0.002 | 0.916±0.009 | 0.847±0.002 |
| LightGBM | 0.840±0.024 | 0.930±0.009 | 0.810±0.000 | 0.926±0.000 | 0.972±0.003 | 0.765±0.055 | 0.900±0.014 | 0.829±0.002 |
| LightGBM† | 0.856±0.018 | 0.930±0.003 | 0.848±0.001 | 0.927±0.001 | 0.998±0.000 | 0.991±0.002 | 0.909±0.011 | 0.843±0.002 |

† indicates optimized models with tuned hyperparameters. TLE represents Time Limit Exceeded.

**Bold** values indicate the best results. Underlined values represent the second best results. * is our implementation.

Table 2: Test metrics on the regression datasets (normalized MSE), lower is better.

| Model | Appliances Energy (⇓) | Bike Sharing (⇓) | California Housing (⇓) | Wine Quality (⇓) | Song Year (⇓) | News Popularity (⇓) | Abalone (⇓) | Microsoft (⇓) |
|---|---|---|---|---|---|---|---|---|
| LASSO | 1.000±0.000 | 1.073±0.005 | 1.040±0.002 | 1.000±0.001 | 1.000±0.000 | 1.000±0.000 | 1.038±0.005 | 1.000±0.000 |
| LASSO† | 1.000±0.000 | 1.041±0.070 | 1.033±0.016 | 1.000±0.000 | 1.000±0.000 | 1.000±0.000 | 0.943±0.215 | 1.000±0.000 |
| GA$^2$M EBM | 2.954±0.469 | 0.065±0.006 | 0.523±0.016 | 0.605±0.004 | 0.687±0.001 | 0.643±0.001 | 0.506±0.014 | 0.839±0.002 |
| GA$^2$M EBM† | 3.680±2.203 | 0.058±0.003 | 0.464±0.019 | 0.603±0.008 | TLE | 0.643±0.003 | 0.484±0.006 | TLE |
| NODE-GAM | 0.805±0.155 | 0.183±0.001 | 0.275±0.010 | 0.339±0.003 | **0.474±0.001** | **0.362±0.002** | 0.465±0.020 | 0.365±0.001 |
| NODE-GA$^2$M | 1.968±1.220 | 0.119±0.004 | **0.230±0.009** | 0.328±0.007 | 0.475±0.001 | 0.366±0.003 | 0.479±0.017 | **0.363±0.001** |
| SIAN-1* | **0.773±0.152** | 0.174±0.003 | 0.343±0.005 | 0.639±0.002 | 0.706±0.000 | 0.674±0.001 | 0.478±0.001 | 0.883±0.000 |
| SIAN-2* | 0.974±0.014 | **0.054±0.002** | 0.285±0.006 | 0.586±0.002 | 0.698±0.004 | 0.726±0.001 | 0.455±0.006 | 0.881±0.003 |
| SIAN-3* | 1.143±0.122 | **0.054±0.002** | 0.266±0.011 | 0.583±0.001 | 0.696±0.002 | 0.751±0.002 | 0.476±0.035 | 0.882±0.001 |
| SIAN-5* | 1.217±0.212 | 0.058±0.004 | 0.262±0.008 | 0.575±0.019 | 0.696±0.002 | 0.753±0.001 | 0.488±0.035 | 0.882±0.001 |
| MLP | 2.252±0.604 | 0.076±0.004 | 0.337±0.018 | 0.584±0.000 | 0.614±0.001 | 0.676±0.001 | 0.496±0.013 | 0.870±0.001 |
| RF | 3.310±0.387 | 0.143±0.004 | 0.420±0.008 | 0.502±0.009 | 0.726±0.001 | 0.652±0.002 | 0.496±0.015 | 0.821±0.001 |
| RF† | 3.362±0.160 | 0.133±0.001 | 0.413±0.015 | 0.486±0.013 | 0.721±0.001 | 0.642±0.001 | 0.484±0.011 | 0.814±0.004 |
| SVM | 0.948±0.005 | 0.395±0.021 | 0.289±0.002 | 0.604±0.004 | 0.665±0.000 | 0.679±0.002 | **0.443±0.002** | TLE |
| SVM† | 1.522±0.108 | 0.731±0.032 | 0.303±0.005 | 0.589±0.005 | TLE | 0.673±0.021 | 0.446±0.008 | TLE |
| XGB | 6.018±1.423 | 0.072±0.003 | 0.403±0.012 | 0.567±0.016 | 0.683±0.001 | 0.690±0.004 | 0.588±0.020 | 0.811±0.001 |
| XGB† | 2.453±1.163 | 0.068±0.003 | 0.411±0.008 | 0.505±0.019 | 0.621±0.001 | 0.630±0.002 | 0.488±0.008 | 0.799±0.002 |
| CatBoost | 1.604±0.156 | 0.069±0.001 | 0.402±0.013 | 0.588±0.001 | 0.711±0.001 | 0.641±0.001 | 0.506±0.008 | 0.823±0.000 |
| CatBoost† | 1.823±0.351 | 0.062±0.009 | 0.402±0.021 | 0.538±0.026 | 0.640±0.001 | 0.633±0.002 | 0.497±0.015 | 0.810±0.005 |
| LightGBM | 2.650±0.561 | 0.072±0.002 | 0.368±0.010 | 0.574±0.004 | 0.686±0.002 | 0.637±0.003 | 0.521±0.012 | 0.812±0.001 |
| LightGBM† | 2.274±0.291 | 0.065±0.002 | 0.433±0.031 | 0.513±0.005 | 0.634±0.001 | 0.628±0.002 | 0.493±0.009 | 0.806±0.003 |

† indicates optimized models with tuned hyperparameters. TLE represents Time Limit Exceeded.

**Bold** values indicate the best results. Underlined values represent the second best results. * is our implementation.

In Tables 1 and 2, we see all of our results across 8 classification tasks and 8 regression tasks. Some major takeaways include the overall good performance of additive models when compared with the performance of blackbox models on a number of datasets. Especially on regression datasets, neural additive models like NODE-GAM are very often the best performing method. Additionally on several of the classification datasets, we see the NODE-GAM and SIAN still performing comparably to XGB, CatBoost, and Light GBM. As argued, a major reason for this being possible is due to the medium dimensionality of tabular datasets, which is even further amplified by the presence of interpretable input features which are common in tabular data. The three classification datasets with the most samples (Tree Cover, Higgs Boson, and Credit Card Fraud) seem to be the some of the datasets where blackbox approaches are the most competitive. Another outlier to this trend is the Madelon dataset, which is a 2003 challenge dataset described as specifically having higher-order interactions.

## 6 CONCLUSION

We have demonstrated across several instances the value of considering neural additive models with feature interactions in both applied settings and in deep learning theory. Although there is still a lot of

room for developing feature interaction selection algorithms, tightening statistical convergence rates, and describing statistical-computational tradeoffs in deep learning, additive models with interactions are already able to give many insights into a data-centric perspective across an abundance of machine learning and deep learning tasks.

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

# A  EXPERIMENT DETAILS

Table 3: Real-world regression datasets. $n$ is the number of samples, $d$ is the number of features.

| Dataset | $n$ | $d$ |
|---|---|---|
| Microsoft | 1,000,000 | 136 |
| Song Year | 515,345 | 90 |
| Online News Popularity | 39,797 | 48 |
| California Housing | 20,640 | 8 |
| Energy Appliances | 19,735 | 30 |
| Bike Sharing | 17,379 | 13 |
| Wine Quality | 6,497 | 12 |
| Abalone | 4,177 | 8 |

Table 4: Real-world classification datasets. $c$ is the number of class labels, $p$ is the percentage with the positive class label.

| Dataset | $n$ | $d$ | $c$ | $p$ |
|---|---|---|---|---|
| Tree Cover | 581,012 | 12 | 7 | – |
| Eucalpytus | 725 | 19 | 5 | – |
| Higgs Boson | 11,000,000 | 28 | 2 | 53.0% |
| Credit | 284,807 | 30 | 2 | 0.2% |
| Adult Income | 48,842 | 13 | 2 | 24.3% |
| Churn | 7,043 | 19 | 2 | 27.0% |
| Taiwanese Bankruptcy | 6,819 | 94 | 2 | 3.1% |
| Madelon | 4,400 | 500 | 2 | 50.0% |

## A.1  REAL-WORLD DATASET DETAILS

**Regression Datasets**

- Microsoft: Predicting the rated relevance of a search query by the user
- Song Year: Predicting the release year of a song based on its audio features
- Online News Popularity: Predicting the popularity of a news article in terms of the number of shares
- California Housing: Predicting the median housing price for different areas in California
- Energy Appliances: Predicting the electricity demand within a home
- Bike Sharing: Predicting the number of bikes which will be rented within an hour
- Wine Quality: Predicting the rated quality of a wine based off of physical measurements
- Abalone: Predicting the age of an abalone (mollusk) based on some physical measurements

**Binary Classification Datasets**

- Higgs Boson: Distinguish between a signal process and a background process
- Credit: Classify whether there is credit card fraud or not (0 or 1)
- Adult Income: Predict if income exceeds 50,000 USD
- Churn: Predict whether there is customer churn or not (0 or 1)
- Taiwanese Bankruptcy: Predicting bankruptcy (0 or 1)
- Madelon: Predict whether a sample belongs to class -1 or +1

**Multi-Class Classification Datasets**

- Tree Cover: Classify 7 types of forest tree cover
- Eucalpytus: Predict the type of the eucalyptus tree

## A.2  MODEL DETAILS

**XGBoost**

- N Estimators: 100 - 1000 (default: 100)
- Learning Rate: 0.01 - 0.3 (default: 0.3)
- Max Depth: 3 - 15 (default: 6)
- Min Child Weight: 1 - 10 (default: 1)

- Subsample: 0.5 - 1.0 (default: 1.0)
- Colsample Bytree: 0.5 - 1.0 (default: 1.0)
- Gamma: 0 - 5 (default: 0)
- Reg Alpha: 0 - 10 (default: 0)
- Reg Lambda: 0 - 10 (default: 1)
- Use Label Encoder: False (default: False), only for classification tasks
- Eval Metric: logloss (default: logloss), only for classification tasks
- Tree Method: hist (default: auto)
- Device: cuda (default: cpu)
- Random State: 2025 - 2029 (default: 0)

**Random Forest**

- N Estimators: 100 - 1000 (default: 100)
- Max Depth: 5 - 30 (default: None)
- Min Samples Split: 2 - 20 (default: 2)
- Min Samples Leaf: 1 - 10 (default: 1)
- Max Features: sqrt, log2 (default: sqrt)
- Bootstrap: True, False (default: True)
- Max Samples: 0.5 - 1.0 (default: None), if bootstrap is True
- Criterion: gini, squared error (default: gini), gini for classification tasks and squared error for regression tasks
- Random State: 2025 - 2029 (default: 0)

**SVM**

- Kernel: linear, poly, rbf, sigmoid (default: rbf)
- C: 0.1 - 30 (default: 1)
- Gamma: scale, auto (default: scale)
- Epsilon: 0.01 - 0.5 (default: 0.1), only for regression tasks
- Probability: True (default: True), only for classification tasks
- Random State: 2025 - 2029 (default: 0), only for classification tasks

**CatBoost**

- Iterations: 100 - 1000 (default: 100)
- Learning Rate: 0.01 - 0.3 (default: 0.1)
- Depth: 3 - 10 (default: 6)
- L2 Leaf Reg: 1 - 10 (default: 3)
- Subsample: 0.5 - 1.0 (default: 1.0)
- Random Strength: 0 - 10 (default: 1)
- Border Count: 32 - 254 (default: 128)
- Verbose: 0 (default: 0)
- Task Type: GPU (default: GPU)
- Bootstrap Type: Bernoulli (default: Bernoulli)
- Eval Metric: Logloss, MultiClass (default: Logloss, MultiClass), Logloss for binary classification tasks and MultiClass for multiclass classification tasks else None
- Random Seed: 2025 - 2029 (default: 0)

**LightGBM**

- N Estimators: 100 - 1000 (default: 100)
- Learning Rate: 0.01 - 0.3 (default: 0.1)
- Max Depth: 3 - 15 (default: 7)
- Num Leaves: 20 - 150 (default: 31)
- Min Child Samples: 10 - 50 (default: 20)
- Subsample: 0.5 - 1.0 (default: 1.0)
- Colsample Bytree: 0.5 - 1.0 (default: 1.0)
- Reg Alpha: 0 - 10 (default: 0)
- Reg Lambda: 0 - 10 (default: 0)
- Verbose: -1 (default: -1)
- Device Type: cpu (default: cpu)
- Objective: binary, multiclass, regression (default: binary, multiclass, regression), binary for binary classification tasks, multiclass for multiclass classification tasks and regression for regression tasks
- Random State: 2025 - 2029 (default: 0)

**EBM**

- Learning Rate: 0.0025 - 0.2 (default: 0.015)
- Max Bins: 256 - 1024 (default: 1024)
- Max Interaction Bins: 16 - 64 (default: 64)
- Interactions: 0.0 - 0.95 (default: 0.9)
- Outer Bags: 8 - 25 (default: 14)
- Inner Bags: 0 - 25 (default: 0)
- Max Leaves: 2 - 3 (default: 3)
- Early Stopping Tolerance: -0.0001 - 0.0001 (default: 0.00001)
- Objective: log loss (default: log loss), only for classification tasks
- Random State: 2025 - 2029 (default: 0)

**Lasso**

- Alpha: 0.0001 - 1000.0 (default: 1.0), only for regression tasks
- Penalty: l1 (default: l1), only for classification tasks
- Solver: liblinear (default: liblinear), only for classification tasks
- C: 0.0001 - 1000.0 (default: 1.0), only for classification tasks
- Random State: 2025 - 2029 (default: 0)

**Optuna – Hyperparameter Optimization**

- N Trials: 100 (default: None), 100 runs for optimized models
- Direction: minimize, maximize, (default: minimize), minimize for regression tasks, maximize for binary and multiclass classification tasks

**MLP**

- Batch Size: 32 (default: 32)
- Learning Rate: $5 \times 10^{-3}$ (default: $5 \times 10^{-3}$)
- Maximum Steps: $2^{20} \approx 1M$

**Batchwise FIS (Algorithm 2)**

- Max $K$: 1, 2, 3, 5 (default: None), corresponding to SIAN-1, SIAN-2, SIAN-3 and SIAN-5 respectively
- Heredity Cutoff Thresholds ($\tau$): Heredity for each $K$ value (default: None). In the format ($\tau_K$) as detailed below:
  - $\tau_1$: 1.0
  - $\tau_2$: 0.5
  - $\tau_3$: 0.33
  - $\tau_4$: 0.25
  - $\tau_5$: 0.20
- Total Rounds ($T$): 50
- Interactions per Round ($B$): 10

We will take weak heredity to mean $\tau_k = \frac{1}{k}$, moderate heredity to mean $\tau_k = \frac{1}{2}$, and strong heredity to mean $\tau_k = 1$. As above, we default to weak heredity.

**SIAN**

- Max $K$: 1, 2, 3, 5 (default: None), corresponding to SIAN-1, SIAN-2, SIAN-3 and SIAN-5 respectively
- FIS style: batchwise (options: layerwise, batchwise, maximal),
  - layerwise as detailed in Algorithm 1,
  - batchwise as detailed in Algorithm 2,
  - maximal consisting of all the interactions with order less than $K$
- Maximum Interactions: 500
- Batch Size: 32
- Learning Rate: $5 \times 10^{-3}$
- Maximum Steps: $2^{17} \approx 125K$

### A.3 SYNTHETIC DATA

We generate synthetic data according to a multilinear mapping of the input features according to a particular interaction structure $\mathcal{I}^*$. Given a sparsity signature $\vec{s} \in \mathbb{N}_0^d$, which is a list of nonnegative integers describing the number of interactions of each order, we ask for a multilinear function which respects this structure. In other words, $s_k = |\{S : |S| = k\}|$. We take a multilinear mapping to mean a polynomial where each input feature has a maximal degree of one. In the case of discrete inputs, we interpret the multilinear function as the corresponding tensor product. For completely continuous $x \in \mathbb{R}^d$, a multilinear term is of the form $(\prod_{k \in S} x_k)$ whereas for a completely discrete $x \in (\otimes_{k=1}^d \{0,1\}^{I_k})$, a multilinear term is instead of the form $\otimes_{k \in S} x_k$. $I_k$ is the number of discrete possibilities in dimension $k$, corresponding to the onehot embedding.

For the $f^{\text{XOR5}}$ dataset, we reinterpret the binary discrete values as taking values in $\{-1, 1\}$ to allow them to be centered around zero for the discrete case, and utilize a uniform distribution for the continuous case. The collection of interactions corresponding to the function $f^{\text{XOR5}}$ is the set $\mathcal{I}* = \{\{1, 2, 3, 4, 5\}\}$, or potentially its hierarchical closure under the assumption of strong hierarchy (heredity).

In all other cases, we consider the choice of $I_k = 5$ for all $k$ unless otherwise specified. The coefficients here are drawn according to a $\beta_S \sim \mathcal{N}(0, 1)$ distribution and then purified according to GAM theory such that each lower dimensional $T \subseteq S$ has any measurable effect on the output. Utilizing these purified coefficients is how the variance bands are able to be computed exactly for these datasets. Finally, according to the band strengths, these $\beta_S$ coefficients are renormalized such that the variance captured by a GAM1, GAM2, GAM3, etc. is at the correct specified level. In the simplest case of normalizing to $(1.0, 1.0, \dots)$, this is $\tilde{\beta}_i = \beta_i / \sum_{i'} \beta_{i'}^2$, $\tilde{\beta}_{i,j} = \beta_{i,j} / \sum_{i',j'} \beta_{i',j'}^2$, etc. These final purified and normalized coefficients are used in the final multilinear function:

$$f(x) = \sum_{S \in \mathcal{I}^*} \tilde{\beta}_S \cdot \left( \prod_{k \in S} x_k \right) \tag{5}$$

