# OpenReview forum: "Revisiting Feature Interaction Selection in Neural Additive Models"
_ICLR.cc/2026/Conference — Submitted to ICLR 2026_

### Official Review · Reviewer_3oKc · 2025-10-31

**Soundness:** 2
**Presentation:** 3
**Contribution:** 1
**Rating:** 2
**Confidence:** 4

**Summary:**

This paper revisits the use of Neural Additive Models and the feature interaction selection paradigm. The authors introduce a new concept called "medium dimensionality," describing a balance between data complexity and model complexity. They argue this phenomenon explains why additive models perform exceptionally well on tabular datasets. Furthermore, the authors further argue that the additive model framework serves as a valuable tool to unify and provide a data-centric perspective on several major deep learning theories, including double descent, grokking, and the staircase property.

The authors then proceed to also present a key algorithmic and architectural improvements for training these models. The authors do so by proposing a "batchwise " selection algorithm for identifying feature interactions, which they find easier to tune than previous methods. They also incorporate "masked training" to compute interaction scores, significantly accelerating the process. Finally, they introduce a "mixed block sparsity" architecture that balances GPU parallelism and memory requirements, proving to be both faster and more memory-efficient than prior implementations. The paper benchmarks these developments, demonstrating the value of NODE-GAMs for both applied performance and theoretical insight.

**Strengths:**

The paper's main strength lies in its ambitious conceptual setting, which aims to provide a novel and valuable contribution by attempting to create a unifying framework to explain four complex deep learning phenomena, such as double descent and grokking. The authors do so through the relatively simplistic lens of additive models. This is a positive highlight, as it offers a simpler, data-centric explanation for these intricate, and well-studied behaviors in the field.

Beyond this theoretical insight, the paper delivers algorithmic and architectural improvements. It introduces a "batchwise" feature interaction selection algorithm that appears to be an enhancement over the previous layerwise method, offering better performance, faster computation, and simpler tuning with fewer hyperparameters. Furthermore, the proposed "mixed block sparsity" architecture appearingly provides a practical solution as it strikes an effective balance between sequential and parallel approaches to be both faster/more space-efficient.

**Weaknesses:**

I have several significant issues with the paper in its current form.

1. Poor Clarity and Precision in Core Concepts

Firstly, the introduction of key deep learning concepts in Section 2 is quite poorly written and lacks precision.#

On Double Descent: The description of double descent merely mentions "two local minima" but completely omits the most critical components of the phenomenon: the spike in test error at the interpolation threshold and the subsequent increase and decrease in test error as model complexity grows. While many readers may be familiar with the concept, it is crucial for an academic paper to define its terms with clarity and precision.

On Grokking: The explanation for grokking is similarly imprecise. The authors characterize one of the learning phases as "weight decay." This is very vague. It would be more accurate to describe this as a compression or regularization-driven search phase, or at the very least, the authors should explicitly define what they mean by "weight decay" in this specific context.

On Notation: In Section 2.2, the authors fail to introduce their notation properly. A clear example is in Equation (4), where the notation is used without any prior definition, leaving the reader to guess its meaning.

2. Unsubstantiated Claims and Vague Concepts

The paper makes several bold claims that are either unsupported or poorly integrated into the narrative.The authors make a broad and, frankly, rude and unnecessary assertion on line 187 that high-dimensional statistics is "mainly applied to specific domains like biostatistics." This is an inaccurate generalization that dismisses a vast field of research.

The newly introduced concept of "medium dimensionality" feels flat. The paper does not clearly demonstrate how or where this concept is truly used. It would be essential for the authors to clarify which of their settings, experiments, or results hold specifically for medium dimensions and not for high dimensions. As it stands, the term is introduced without a clear purpose or payoff.

3. Weak Experimental Design and Justification

Several of the experiments and claims in Section 3 are unconvincing or poorly justified. In Section 3.1, the authors describe a synthetic dataset but would be better served by explicitly and formally defining what this dataset is rather than just describing it in prose. In Section 3.2, the authors state, "even if we know the true model, it may not be the best to fit with finite data." This is a very bold claim to include without solid proof, clarification, and strong experimental backing. The provided experiments are in my opinion too weak to support such a general statement.

The experiment in Section 3.3 is also problematic. The authors introduce an uncommon multiplicative setting to demonstrate grokking. The choice of this dataset and the $y \leftrightarrow x$ relationship seems arbitrary. The results are not convincing, as they only appear to show the grokking phenomenon for a single step

4. Structural and Narrative Flaws

The paper suffers from a lack of cohesion, in my opinion it currently reads a bit like a collection of disparate parts. Section 4 feels completely disconnected from the rest of the paper. After a theoretical setup in the preceding sections, this section on algorithmic improvements arrives "out of nowhere", ruining the flow of the paper.

The results in Section 5 (Tables 1 and 2) are concerning. In several instances, the optimized models with tuned hyperparameters perform worse than their untuned counterparts. For example, in Table 2, the tuned GA2M EBM has a lower normalized MSE than its counterpart, and the same occurs for the SVM on the Appliances Energy dataset. This pattern repeats across both tables and multiple datasets/methods, undermining the claims about the proposed optimizations. If i misundsterstood this, I would greatly appreciate if the authors could clarify it.

Finally, the conclusion's mention of "tightening statistical convergence rates" also seems to appear from nowhere and is not linked to any of the paper's actual content or analysis. In general, while the theme of the paper is additive models, it feels like a bundle of many small, underdeveloped analyses. Unfortunately, I do not believe any of these analyses are independently sufficient for acceptance, and their links are too weak. I strongly suggest the authors focus on building a better narrative and either thoroughly corroborate the claims I pointed out or remove them entirely.

**Questions:**

Please see weaknesses.

---

> ### Author Response · Authors · 2025-11-14
>
> Hello Reviewer 3oKc,
>
> Thank you for your review.  We are commenting to ask if it is possible for you to remove the LLM-generated portions from your review.  Unfortunately, we find it difficult to understand your intention and it is making it difficult for us to begin an engaging discussion.  Currently, there are several errors in the review and it seems like some of your meaning could have been lost during AI paraphrasing.  Can you remove these issues as soon as possible so that we would have a fair chance to respond before the November 20th deadline?  We believe it would be easier for us to address/incoporate your feedback into improving our paper if you condense down to your main thoughts and main questions.  Thank you for your help.

---

> > ### Comment · Reviewer_3oKc · 2025-11-14
> >
> > Dear Authors,
> >
> > Thank you for your response. Although I find your accusal of using LLMs to generate my review, which I have spent a significant amount of time in writing, disappointing, below I provide the four main points and weaknesses that I have found within the main manuscript:
> > 1. The concepts such as Double Descent and Grokking are poorly introduced. Please rewrite the corresponding paragraphs to include a more precise description.
> > 2. The "medium dimensionality" seems too broad. Please either provide stronger justification or remove it.
> > 3. The paper is not cohesive. Please improve justification of Section 4 and how it relates to previous parts.
> > 4. There are numerous inconsistencies in your Figures and Tables (which reviewer QFjS pointed out as well). Please address these.
> >
> > Finally, if there are any parts of my previous review which you did not understand or are unclear, I would be more than happy to clarify them if you could point out exactly what was unclear.

---

> > > ### Author Response · Authors · 2025-11-14
> > >
> > > Hello Reviewer 3oKc,
> > >
> > > Thank you for your quick response.  We do not mean to make you feel accused.  We will proceed under the assumption that your review was completely written by you and will address each point you made accordingly.  We will later also try to ask for further clarification on the parts which were unclear to us.  Thank you in addition for your condensed version of the weaknesses you have discovered in the manuscript.

---

> ### Author Response · Authors · 2025-11-22
>
> We will begin with responding to your second review because we generally find it easier to understand.
>
> > The concepts such as Double Descent and Grokking are poorly introduced
>
> We are sorry that the brief paragraphs in Section 2 were not enough for you to understand these well-studied phenomena.  Unfortunately, it is always a balance of writing sufficient detail so that the uniformed reader can get some idea of what is going on without taking up too much space for the well-acquainted reader.  If you have any questions on this which are still unclear or you have some specific suggestions on how to better write this, please let us know.  Regarding your original suggestions on these in your longer review, we will discuss these later.
>
> > The "medium dimensionality" seems too broad. Please either provide stronger justification or remove it.
>
> Unfortunately, the concept of medium dimensionality is a central theme of the paper and there is absolutely no possibility of removing it.  On the contrary, it is specifically designed to be broad in covering the region between classical statistics (low-dimensional) and high-dimensional statistics.  We hope the other points we make will also help strengthen the justification of why medium-dimensionality is necessary.
>
> We show its theoretical importance for demonstrating the weaknesses in our current understanding of deep learning theory and we show its practical importance for achieving state-of-the-art performance on tabular datasets.  It is unclear to us what the reviewer has in mind as a stronger justification than these.
>
> > The paper is not cohesive. Please improve [the] justification of Section 4 and how it relates to [the] previous parts.
>
> We apologize that Section 4 did not feel properly introduced.  To reemphasize its purpose, our major claim in Section 3 is how additive models are necessary to understand the medium dimensionality explored, and so for the reader’s benefit, we spend Section 4 to work on the practical usefulness of those same additive models.  As you say yourself in your review: “Beyond this theoretical insight, the paper delivers algorithmic and architectural improvements.”  We have added a paragraph to the beginning of the section to make this transition easier.
>
> > There are numerous inconsistencies in your Figures and Tables…
>
> We will address the *weaknesses* in the figures pointed out by Reviewer QFjS in our response to them, regarding “inconsistencies” we are not sure what you mean.  Regarding the issues in the tables, we only see reference to your first response where you indicate that sometimes the tuned models perform worse than the untuned models.  As a reminder, you point out that this happens at least 2 times out of the total possible 7*16=112 possible locations.  It is unclear what you are suggesting as the inconsistency here.  We believe the only possible interpretation is that you are suggesting we should tune the hyperparameters directly on the test dataset.  Unfortunately, we strongly believe this is ill-advised.  If we misunderstood this somehow, we would greatly appreciate it if the reviewer could clarify what their implication is.

---

> > ### Author Response · Authors · 2025-11-22
> >
> > Now, addressing your original, longer review and the several points of confusion we were left with.
> >
> > > The description of double descent merely mentions "two local minima" but completely omits the most critical components of the phenomenon: the spike in test error at the interpolation threshold and the subsequent increase and decrease in test error as model complexity grows.
> >
> > In the same paragraph we discuss the variance spike occurring at the interpolation point.  We do not feel this is necessarily the most critical component as it does not even occur in all settings.
> >
> > > On Grokking: … The authors characterize one of the learning phases as "weight decay." This is very vague. It would be more accurate to describe this as a compression or regularization-driven search phase, or at the very least, the authors should explicitly define what they mean by "weight decay" in this specific context.
> >
> > The term “weight decay” occurs directly next to a citation which should explain the suggested phenomenon in greater detail.  To be clear, “weight decay” refers to the technical term for the method used to shrink weights in most gradient descent algorithms.
> >
> > > …fail to introduce their notation properly. A clear example is in Equation (4), where the notation is used without any prior definition, leaving the reader to guess its meaning.
> >
> > Please check the notation in the first paragraph of Section 2.  $x_{-s_1}$ means the set of all features except for the feature $s_1$.
> >
> > > The authors make a broad and, frankly, rude and unnecessary assertion on line 187 that high-dimensional statistics is "mainly applied to specific domains like biostatistics."
> >
> > In the original context of *“Unlike high-dimensional statistics, which is mainly applied to specific domains like biostatistics (where d ≫ n often holds), feature interaction selection is more widely applicable across machine learning tasks (since $2^d$ ≫ n more commonly holds)”*, we find it extremely difficult to understand why you find the assertion rude or dismissive.  If you could provide further details on this or suggestions for alternatives, please let us know.
> >
> > > The newly introduced concept of "medium dimensionality" feels flat. The paper does not clearly demonstrate how or where this concept is truly used. … As it stands, the term is introduced without a clear purpose or payoff.
> >
> > As mentioned, this newly introduced concept is a central theme of our paper.  Perhaps if you are an expert in high-dimensional statistics, you can explain how those existing results directly apply to our setting.  Otherwise, currently, we strongly feel experiments like in Figure 4 are representative of the rich transition from low-dimensional behavior to high-dimensional behavior.
> >
> > > In Section 3.2, the authors state, "even if we know the true model, it may not be the best to fit with finite data." This is a very bold claim to include without solid proof, clarification, and strong experimental backing
> >
> > We believe this statement is a widespread belief within the community of high-dimensional statistics.  For example, in Figure 2, we can see how this is clearly true in the range of 200 to 2000 samples.
> >
> > > The authors introduce an uncommon multiplicative setting to demonstrate grokking
> >
> > Are you arguing that we did not show grokking or that our dataset choice is not possible for showing grokking?
> >
> > > Finally, the conclusion's mention of "tightening statistical convergence rates" also seems to appear from nowhere and is not linked to any of the paper's actual content or analysis
> >
> > We discuss this in the final conclusion section of the work.  As is mentioned throughout, the focus is on the data-centric and statistical arguments allowed by additive models.  We feel tightening statistical results is a natural direction of future development, even if it does not directly match our analysis.
> >
> > > I strongly suggest the authors focus on building a better narrative and either thoroughly corroborate the claims I pointed out or remove them entirely.
> >
> > Can the reviewer please make explicit which are the “claims” from their long first review which we should focus on corroborating?  This would help us to understand what aspects of the paper you felt were and were not properly corroborated.

---

> > > ### Comment · Reviewer_3oKc · 2025-11-25
> > >
> > > I thank the authors for their detailed response.
> > >
> > > As the authors correctly point out, it is indeed always a 'balance of writing sufficient detail such that the uninformed reader can get idea of whats going on', which is precisely why I believe the paper needs to be rewritten. The concepts need clearer and better explanations.
> > >
> > > > We show its theoretical importance for demonstrating the weaknesses in our current understanding of deep learning theory and we show its practical importance for achieving state-of-the-art performance on tabular datasets. It is unclear to us what the reviewer has in mind as a stronger justification than these.
> > >
> > > Regarding 'medium dimensionality' I personally was not convinced of the arguments the authors gave and still believe the concept is too broad.
> > >
> > > > It is unclear what you are suggesting as the inconsistency here. We believe the only possible interpretation is that you are suggesting we should tune the hyperparameters directly on the test dataset. Unfortunately, we strongly believe this is ill-advised. If we misunderstood this somehow, we would greatly appreciate it if the reviewer could clarify what their implication is.
> > >
> > > My original intention was just to mention that I was rather surprised by how many times did the tuned models performed worse than the untuned models.
> > >
> > > > In the original context of “Unlike high-dimensional statistics, which is mainly applied to specific domains like biostatistics (where d ≫ n often holds), feature interaction selection is more widely applicable across machine learning tasks (since $2^d$ ≫ n more commonly holds)”, we find it extremely difficult to understand why you find the assertion rude or dismissive. If you could provide further details on this or suggestions for alternatives, please let us know.
> > >
> > > The statement seems to imply that high-dimensional statistics is "mainly" applied to biostatistics. I think the word choice of "mainly" here is poor.
> > >
> > > > Are you arguing that we did not show grokking or that our dataset choice is not possible for showing grokking?
> > >
> > > I am saying that the example you gave seems rather uncommon. If you could give further examples/different kinds (not just multiplicative) I think it would give a clearer/stronger message.
> > >
> > > Overall, I still stand with my initial assessment.

---

> > > > ### Author Response · Authors · 2025-11-26
> > > >
> > > > Hello Reviewer 3oKc,
> > > >
> > > > We are happy to address your final response, but could you first address all of the points we made?
> > > >
> > > > You previously said “if there are any parts of my previous review which you did not understand or are unclear, I would be more than happy to clarify them if you could point out exactly what was unclear.”  We feel there are several points of confusion which you have still not addressed:
> > > >
> > > > - double descent merely mentions "two local minima"
> > > > - "weight decay" is very vague
> > > > - missing notation in Equation (4)
> > > > - can the reviewer provide a little more specificity on ‘medium dimensionality’ *not having a clear purpose or payoff*?
> > > > - "even if we know the true model, it may not be the best to fit with finite data." does the reviewer still disagree with this statement?
> > > > - which are the “claims” which the reviewer believes are not corroborated?

---

### Official Review · Reviewer_QFjS · 2025-11-01

**Soundness:** 2
**Presentation:** 1
**Contribution:** 2
**Rating:** 2
**Confidence:** 3

**Summary:**

this paper analyzes a few deep learning phenomena with NAM (neural network generalized additive models). The authors focused on the set of medium dimensionality where 2^d >> n (d is the feature dimension, n is the number of training data) and reproduced a few phenomena (double descent, staircase dynamics. From these observations, the authors proposed batchwise feature interaction selection/masked training/mixed block sparsity for improving NAM.

**Strengths:**

* this paper shows some lead that one can improve the NAM by leveraging known properties of deep learning training dynamics
* focusing on medium dimensionity setup is useful because many real world problems fall into this category

**Weaknesses:**

the paper presentation is pretty bad:
* figure 2 is not readable: all subtitle are the same,
* figure 3 is hard to understand
* figure 6 is much hard to reader than that if you just present a table


on the other side, the connection from section 3 to section 4 can be significantly improved. it's probably the most important part (where you observe somethings and then adjust the methodology accordingly) so I would suggest spend more sentences/paragraphs to highlight how and why you made the proposals on section 4 based on what you found on section 3.

**Questions:**

questions about medium dimensionality
1. it's a quite wide spectrum between [d, 2^d]. does things happen in the whole spectrum or there are some more precision criterion?
2. is the medium dimensionality feature for data, or for model, or for both together?

---

> ### Author Response · Authors · 2025-11-22
>
> Thank you for your review, we will address each of your points sequentially.
>
> > S1. this paper shows some lead that one can improve the NAM by leveraging known properties of deep learning training dynamics
>
> That is not quite correct, we discover a novel property of NAM training dynamics which we attribute to the effects of **medium dimensionality**.  The explanation of these deep learning phenomena through the data-centric and statistical lens are not well-studied previously.  Double descent is often viewed from a model parameters perspective.  Grokking is viewed from a training time perspective.  Staircase works make the assumption of infinite training data.  We believe our work provides a unique insight here using NAMs as a tool for data-centric insight.
>
>
> > S2. focusing on medium dimensionity setup is useful because many real world problems fall into this category
>
>
> That’s exactly right, we extend to a medium dimensional setting which is much more realistic for many real-world settings.  We show how existing works leave a gap by not focusing on this, despite how effective it is at describing neural network behavior.
>
> > W1. figure 2 is not readable: all subtitle are the same,
>
> You are correct that all subtitles were the same representing that the dataset remained the same.  We have now labeled 2b,c,d with the number of epochs trained and moved the original subtitle to the caption.  Please consider reading the caption of Figure 2 for additional details.  Each figure plots the test error of three models with different K values.  These figures demonstrate both double descent and medium dimensionality.  If you have any specific questions about Figure 2 remaining, please ask us directly.
>
> > W2. figure 3 is hard to understand
>
> Please consider reading the grokking paper [C] if you are still having trouble understanding Figure 3.  In orange we plot the ‘high sample’ curves where train and val error match closely.  In black, we plot the ‘low sample’ curves where train and val error diverge and exhibit grokking behavior.  Sweeping between the N value (number of samples) shows a smooth transition between these behaviors as a function of the dataset size.
>
> > W3. figure 6 is much hard to reader than that if you just present a table
>
> Unfortunately, we disagree with you that a table would more succinctly represent the provided information.  We believe that the main takeaways –that smaller K values are faster and that our introduced masking approach is faster –are represented clearly using the bar chart.
>
> > W4. on the other side, the connection from section 3 to section 4 can be significantly improved. it's probably the most important part (where you observe somethings and then adjust the methodology accordingly) so I would suggest spend more sentences/paragraphs to highlight how and why you made the proposals on section 4 based on what you found on section 3.
>
> We are sorry that the writing makes you think Section 3 directly affects Section 4.  Actually, Sections 3 and 4 are two independent contributions (empirical theory vs practical developments).  The importance of NAMs in Section 3 (theory) and in Section 5 (real world) both inspire the need for the speedups and ease-of-use in Section 4, but we cannot move Section 4 earlier because of the need to motivate NAMs in Section 3.  We have added paragraphs to the end of Section 1 and beginning of Section 4 to make these transitions smoother.
>
>
> > Q1. it's a quite wide spectrum between [d, 2^d]. does things happen in the whole spectrum or there are some more precision criterion?
>
> Yes, that’s right.  It is a very wide spectrum between [d, 2^d] which is why our work tries to focus on the medium-dimensional setting which is not covered by existing low-dimensional statistics (N>2^d) or high-dimensional statistics (N<d).  There are many intermediate behaviors occurring in this range.  Accordingly, that is why we decided to plot the entire spectrum in Figure 2.  Recall for this figure that d=10 means that [d,2^d] ~= [10,1000].
>
> > Q2. is the medium dimensionality feature for data, or for model, or for both together?
>
> We use medium dimensionality to describe a wider phenomenon exactly like high dimensionality.  In particular, it is a property of the data and the model **together**. Please also see the footnote on page 4.
>
> Looking at Figure 2, you can see that changing the x-axis (N value) changes the data and changing the K value changes the model.  Both of these changes affect the learning behavior and the test performance **together**.  This is the key insight this paper is trying to focus on.
>
> [C] “Grokking: Generalization Beyond Overfitting on Small Algorithmic Datasets”. Alethea Power et al. 2022.

---

### Official Review · Reviewer_JAkH · 2025-11-03

**Soundness:** 2
**Presentation:** 1
**Contribution:** 2
**Rating:** 2
**Confidence:** 4

**Summary:**

This paper tackles the problem of feature interaction selection for neural additive models. The authors first review existing deep learning theory and empirical observations of training dynamics. Then, they experimentally show that the relationship between data complexity and model complexity helps to explain the good performance of additive models. Additionally, they demonstrate that the phenomenon of deep learning theory can be observed in additive models. Finally, the authors propose a feature interaction selection method by modifying the existing SIAN method, and apply it to several tabular datasets.

**Strengths:**

- This paper focuses on the deep learning phenomena, such as double descent and grokking, on neural additive models. The authors experimentally show that these phenomena can be observed in neural additive models.
- The authors propose a feature interaction selection method by modifying the existing method, SIAN. The proposed method introduces a batch-wise selection and masked training to SIAN. In addition, the authors employ a mixed block-wise sparsity approach to accelerate the computational time of neural additive models.

**Weaknesses:**

- The presentation of this paper is not well. The relationship between empirical observations of additive models presented in Section 3 and the algorithm development in Section 4 is not clear.
- The experimental results do not support a clear advantage of the proposed method. The reviewer assumes that the SIAN-X in Tables 1 and 2 refers to the proposed method. There is no comparison between the original SIAN and the proposed method.
- The ablation study is missing to show the effectiveness of each component of the proposed method, batchwise selection, and masked training.
- The proposed method seems an incremental extension of the existing SIAN. The novelty and significance of the proposed method are limited.
- The detailed experimental settings, such as the hyperparameter settings for SIAN, are missing.

**Questions:**

- The authors should clarify how the observations in Section 3 lead to the proposed method in Section 4.
- How is the contribution of each component in the proposed method for the performance?? An ablation study is necessary to clarify this point.
- How do the authors determine the hyperparameters for SIAN and the proposed method? ($K$, $B$, $T$, etc.)
- The following literature tackles a feature interaction selection for neural additive models. In addition, the efficient implementation of neural additive models is mentioned, which is related to Section 4.3 of this paper.

Kishimoto, Y., et al., Neural Additive and Basis Models with Feature Selection and Interactions, PAKDD 2024, https://doi.org/10.1007/978-981-97-2259-4_1

---

> ### Author Response · Authors · 2025-11-22
>
> ## Existing Phenomena
>
> > This paper focuses on the deep learning phenomena, such as double descent and grokking, on neural additive models. The authors experimentally show that these phenomena can be observed in neural additive models.
>
> We would like to begin our response by quickly clarifying that we believe our work is not only about experimentally demonstrating these existing deep learning phenomena, but about the unified lens through which all of these separate phenomena can be understood.  In particular, we focus on showing how a known phenomenon occurring in a neural additive model demonstrates the data-centric insight of statistical complexity and how the problem given by feature interaction selection (the collection $\cal{I}$) provides a nonparametric notion of complexity for understanding neural network behavior.
>
> More specifically, double descent is typically viewed in terms of increasing model parameters and grokking is typically viewed in terms of increasing epochs.  We can instead understand both in terms of a mismatch in dataset complexity and modeling complexity.  Further, works on the staircase phenomenon usually assume the presence of infinite training data.  Using NAMs, we can see how this behavior continuously deteriorates with respect to the amount of available data.
>
> We hope this perspective will continue to become more clear as we address your individual questions below.
>
>
> ## Missing Experiments
>
> > How is the contribution of each component in the proposed method for the performance?? An ablation study is necessary to clarify this point.
>
> > The ablation study is missing to show the effectiveness of each component of the proposed method, batchwise selection, and masked training.
>
> In Section 4, we run multiple ablation studies to test the improvements due to our suggested components.  In Figure 6, one can see a \~100x improvement in FIS algorithm speed by utilizing the proposed masking approach.  Because function evaluations are the major component of the FIS algorithm calling to importance estimates, using the much faster masking approach compared to the original method leads to significant speedups.  In Figure 8, one can see a 3\~10x improvement in SIAN training speed utilizing our new block sparsity approach.  For a large number of interactions, the original SIAN approach becomes too large for the GPU and will be completely unusable.  Finally, the original FIS algorithm has some extremely difficult to tune hyperparameters which make it hard to adjust to new datasets.
>
> On a potential point of confusion, some of your questions seem to imply that you would like this evaluation done in terms of the *prediction performance*.  Is this your original intention?  Our goal in Section 4 is to show improvements on the ease of use and training speed for SIAN models, rather than improve the prediction performance of the algorithm.  We hope this also partially addresses your question of why we did not directly compare in Tables 1 and 2 against the original SIAN.  In this work, we do not attempt to improve on the FIS+NAM structure introduced by the SIAN paper, but instead to revisit FIS and the various practical challenges which are actually faced by their algorithm.
>
> > The proposed method seems an incremental extension of the existing SIAN. The novelty and significance of the proposed method are limited.
>
> We believe that you are seriously underestimating the importance of the changes we make.  For around 300 interactions, the training is 10x faster than the vanilla SIAN approach and 3x faster than the blocksparse SIAN approach.  For 1000 interactions, the training is 10x faster than vanilla and ∞x faster than the blocksparse approach.  The choice of batchwise selection of feature interactions is only a minor modification to the original layerwise approach; however, the batchwise approach is not mentioned anywhere in the SIAN paper, and we find makes a significant difference in the ease of use for the FIS algorithm.  When cutting off to 500 interactions like we do, exploring the {20 choose 5} = 15K interactions of order 5 in a batchwise (DFS) way instead of a layerwise (BFS) way makes a significant difference in the efficiency of the exploration.

---

> > ### Author Response · Authors · 2025-11-22
> >
> > > The experimental results do not support a clear advantage of the proposed method. The reviewer assumes that the SIAN-X in Tables 1 and 2 refers to the proposed method. There is no comparison between the original SIAN and the proposed method.
> >
> > We understand that this means you want a comparison with the original SIAN in terms of prediction performance like in Tables 1 and 2.  We hope we have been able to justify why we did not feel like such a comparison was necessary, but, to reiterate, our improvements on the SIAN algorithm in Section 4 were mainly targeting the ease-of-use and speed of the FIS algorithm.  Because we use a very similar set of architecture parameters to the original SIAN method, we believe that if the original FIS and our FIS were to return the same collection of interactions, they would show very similar prediction performance on the real-world dataset.  Accordingly, most of the clear advantages over the original SIAN are with respect to time comparisons.  Regarding these time comparisons, we feel that these comparisons have been addressed in Section 4, even if they do not explicitly state they are a comparison between the original SIAN and our work.
> >
> >
> > ## Writing Concerns
> >
> > > The authors should clarify how the observations in Section 3 lead to the proposed method in Section 4.
> >
> > Section 3 is about the empirical theory contributions by the neural additive model.  Section 4 is about our practical contributions to the neural additive model.  Section 3 **does not** directly “inspire” the choices made in Section 4.  Section 3 demonstrates a need to study NAMs for DLT.  Section 4 improves the current usage.  Section 5 demonstrates the practical importance.  We have added a paragraph at the end of §1 and the beginning of §4 to clarify this.
> >
> > > How do the authors determine the hyperparameters for SIAN and the proposed method? (K, B, T, etc.)
> >
> > > hyperparameter settings for SIAN, are missing.
> >
> > As can be seen in Tables 1 and 2, K is chosen based on the dataset.  B is equal to 10 and T is equal to 50, leading to a maximum of 500 interactions. We have added the several missing hyperparameters to the appendix.
> >
> >
> > ## Related Work
> >
> > > The following literature tackles a feature interaction selection for neural additive models. In addition, the efficient implementation of neural additive models is mentioned, which is related to Section 4.3 of this paper.
> >
> > We first note that the paper you share [A] is only considering interaction pair selection, not the more general higher-order feature interaction selection which we consider.  This distinction was addressed in the last two paragraphs of our §2.
> >
> > Regarding the topic of efficient implementation related to our §4.3, you are correct that it is mentioned in §3.3 of [A].  Correct us if we are wrong, but this paper only mentions comparing to the 1D convolution approach of the Neural Basis Model (NBM) paper [B].  This is not relevant for our findings which instead focus on the typical setting where each feature interaction can learn a unique shape function.  They provide no code to enable any further comparison or confirmation of this.
> >
> > [A] Kishimoto, Y., et al., Neural Additive and Basis Models with Feature Selection and Interactions, PAKDD 2024
> >
> > [B] Filip Radenovic et al. “Neural Basis Models for Interpretability”. NeurIPS 2022.

---

### Meta-Review · Area_Chair_srsD · 2025-12-13

**Summary:**

Concerns are primarily about clarity, cohesion, and positioning of the paper’s contributions. All three reviewers raised issues regarding presentation quality and narrative structure. Two reviewers questioned the novelty and strength of the algorithmic contributions relative to prior work. The concept of "medium dimensionality" was viewed as overly broad, with doubts about its concrete payoff. While the paper was recognized as ambitious and potentially insightful, reviewers were not convinced that the theoretical claims and experimental evidence were sufficiently well-motivated, clearly articulated, or strongly supported to meet the bar for acceptance.

There are couple of other things to note:
- The overall review quality for the paper is below average. However, some of the points made by the reviewer are valid.
- Authors should especially consider framing the story differently or be more explicitly. This is a major revision and note within the scope of the iclr rebuttal.
- Various meta-level decisions by authors could have be done better (suboptimal choice of primary area, communication with reviewers, the rebuttal, etc.).

Given these points, I am very certain that authors would not have been able to change the reviewers' assessment. This might have been to some extent the result of this specific reviewer assignment, but how the paper is positioned certainly influences this assignment. Hence, criticism of the paper should be the primary thing authors should focus on before resubmitting.

**Reviewer Concerns:**

Addressed:

- Missing experimental details and hyperparameter specifications
- The purpose and scope of the algorithmic contributions
- Ablation-style evidence for speedups and practical improvements
- Several presentation issues

Concerns that remain outstanding:

- The narrative flow of the paper
- Criticisim regarding "medium dimensionality"
- Clarity
- Presentation quality

**Reviewer Scores:**

- Reviewer JAkH: Likely unchanged. While several factual concerns (missing details, ablations, hyperparameters) were addressed, the reviewer’s core reservations about novelty, significance, and presentation would likely persist, keeping the score at reject.
- Reviewer QFjS: Might improve slightly in confidence but unlikely to change the overall score. The authors clarified the intent of Sections 3 and 4 and addressed some figure-related issues, but the reviewer’s main concerns about presentation quality and conceptual linkage remain only partially resolved.
- Reviewer 3oKc: No change.

---

### Decision · Program_Chairs · 2026-01-26

Reject